# QED at NNLO with McMule

**P. Banerjee[1], T. Engel[1,2], A. Signer[1,2][⋆], and Y. Ulrich[1,2]**

**1** Paul Scherrer Institut, CH-5232 Villigen PSI, Switzerland
**2** Physik-Institut, Universität Zürich, Winterthurerstrasse 190, CH-8057 Zürich, Switzerland

⋆ adrian.signer@psi.ch

## Abstract

McMule is a framework for fully differential higher-order QED calculations of scattering and decay processes involving leptons. It keeps finite lepton masses, which regularises collinear singularities. Soft singularities are treated with dimensional regularisation and using FKS$^\ell$ subtraction. We describe the implementation of the framework in Fortran 95, list the processes that are currently implemented, and give instructions on how to run the code. In addition, we present new phenomenological results for muon-electron scattering and lepton-proton scattering, including the dominant NNLO corrections. While the applications presented focus on MUonE, MUSE, and P2, the code can be used for a large number of planned and running experiments.

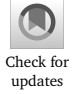

# 1   Introduction

Perturbation theory is a well-established tool to provide accurate theoretical descriptions of many scattering and decay processes. In fact, it is often the case that the coupling (either electromagnetic, strong, or electroweak) is small enough to facilitate a perturbative treatment and non-perturbative effects are either subdominant or can be isolated and modelled to a sufficient precision. Hence, there has been a huge effort and impressive progress in computational techniques for higher-order perturbative calculations.

While most of the effort of the community is geared towards high-energy colliders, there is also a very important low-energy programme ongoing. For example elastic electron-proton scattering at the Jefferson Laboratory lead to a determination of the weak charge of the proton by QWeak [1] or allowed PRad [2] to provide crucial input towards the solution of the proton radius puzzle [3,4]. The same process has been measured at MAMI by the A1 collaboration [5] to determine form factors and will be studied again at MESA, where P2 [6] aims at a precise determination of the weak mixing angle through an asymmetry measurement at a small beam energy of 155 MeV. A similar approach but using electron-electron scattering is pursued by the Moller experiment [7].

Two planned experiments for which we provide new results are MUonE [8] and MUSE [9]. The idea of MUonE is to use a 150 GeV muon beam at CERN to measure the differential cross section for elastic muon-electron scattering at a centre-of-mass energy of $\sqrt{s} \sim 400$ MeV. This is motivated by the connection [10] of hadronic vacuum polarisation (HVP) effects with the anomalous magnetic moment of leptons. From the shape of the muon-electron cross section it is possible to extract the effective electromagnetic coupling and, hence, to obtain an independent determination of the leading hadronic contribution. The idea of MUSE is to measure simultaneously electron-proton and muon-proton scattering, for positively and negatively charged leptons. The experiment will be carried out at the Paul Scherrer Institut with lepton momenta $\mathcal{O}(100\,\text{MeV})$ and will shed further light on the proton radius puzzle and two-photon exchange contributions.

Bhabha scattering is a further example which has been studied extensively [11] in connection with luminosity measurements. Finally, we mention muon and tau decay processes that can be described through QED corrections in the Fermi theory. This list is tailored towards the applications discussed in this paper and is by no means complete. But it shows that there is a demand for precise higher-order QED calculations for low-energy scattering and decay processes involving leptons. It is the aim of McMule (Monte Carlo for MUons and other LEptons) to provide a Monte Carlo code that can be used to obtain precise theoretical predictions for a wide range of low-energy processes dominated by QED effects, with a particular focus on processes involving muons. More precisely, McMule is an integrator that allows to obtain histograms for arbitrary, fully differential observables.

QED calculations are typically simpler than QCD computations. First, due to the abelian nature of QED, the algebra is less involved. A more important aspect is the simplified structure of infrared singularities in QED, which reduces the complexity of the divergent phase-space integrations. Generally, it is a highly non-trivial problem to move from matrix elements to fully differential physical observables. However, the abelian gauge structure of QED leads to a simple Yennie-Frautschi-Suura (YFS) exponentiation of multiple soft singularities [12]. Also, in QED collinear singularities are only possible if a gauge boson (photon) becomes collinear to a fermion. These singularities can be regularised through non-vanishing fermion masses.

The relative simplicity of QED might well be responsible for a remarkable divide in the computational techniques that are used in the QED and QCD community. Typically, scattering processes in QED are computed using an infinitesimal photon mass to regularise infrared singularities and using a slicing method to extract the infrared-divergent part of phase-space

integrations. In MCMULE we follow more closely techniques familiar from QCD calculations and therefore use dimensional regularisation and a subtraction scheme. In this context, the simplicity of the infrared structure of QED has been exploited in [13], where a subtraction scheme at next-to-next-to leading order (NNLO) and beyond has been developed that allows to obtain arbitrary fully exclusive quantities as soon as the matrix elements are known.

Despite the simplicity of QED, there is one aspect in which QED computations are more complicated than QCD calculations. It is related to potentially large logarithms $\log(m^2/Q^2)$ that are remnants of collinear singularities. Here, $Q$ is a typical scale of the process, which is often much larger than some of the fermion masses $m$. In QCD, quantities are usually considered that are inclusive enough such that final-state collinear singularities cancel. Hence, no corresponding large logarithms appear in the final result. Initial-state collinear singularities are factorised into parton distribution functions. Thus, it is possible to set $m = 0$, often to a very good approximation. In QED, this is not the case. Many distributions that are measured are dominated by these logarithms, such that it is often not possible to work with massless leptons. The dominance of the logarithmic terms can be exploited to obtain approximate expressions for higher-order corrections, see e.g. [14] for a review. Keeping finite fermion masses is a substantial complication for the evaluation of virtual corrections. In addition it potentially leads to numerical problems if a fully differential Monte Carlo approach is taken. Thus, in many cases QED results cannot be simply extracted from corresponding QCD results, but a dedicated effort is required.

The Fortran 95 code MCMULE can be downloaded at https://gitlab.com/mule-tools/mcmule, where also an up-to-date table of implemented processes, a documentation, and some sample results can be found. At the time of writing, the following processes are implemented:

$$
\begin{aligned}
&\ell \to \ell' \nu \bar{\nu} && \text{NNLO} \\
&\ell \to \ell' \nu \bar{\nu} \gamma && \text{NLO} \\
&\ell \to \ell' \nu \bar{\nu} (l^+ l^-) && \text{NLO} \\
&\ell p \to \ell p && \text{NLO and dominant NNLO} \\
&\ell \ell' \to \ell \ell' && \text{NLO and dominant NNLO,}
\end{aligned}
\tag{1}
$$

where $\ell$ and $\ell'$ are different leptons and $l$ is either equal to $\ell'$ or the third possible lepton. The lepton decay processes are computed in the Fermi theory. For the processes with a proton $p$ the approximation is made whereby its interaction is only due to the exchange of a single photon.

In this article we will start in Section 2 by briefly recapitulating the techniques we use to do fully differential higher-order QED calculations. The structure of the code, which consists of several modules with a simple, mostly hierarchic structure is described in Section 3. In Section 4 we perform a basic leading-order (LO) calculation in order to illustrate how to run the code. The following two sections are devoted to our main new phenomenological results. We start with MUonE in Section 5. First we explain how to use MCMULE to reproduce next-to-leading order (NLO) results available in the literature [15]. Then we present new results for $\mu$-$e$ scattering, including numerically dominant NNLO corrections. Section 6 is devoted to lepton-proton scattering. We discuss how to extend the partial NNLO calculation of the previous section to elastic $e$-$p$ and $\mu$-$p$ scattering and provide some phenomenological results adapted to P2 and MUSE. These processes are just the beginning of the MCMULE programme. In Section 7 we discuss possible future developments of MCMULE. Finally, the input parameters used by MCMULE are listed in Appendix A.

## 2  QED corrections as implemented in MᴄMᴜʟᴇ

As mentioned in the introduction, some of the techniques used within MᴄMᴜʟᴇ are somewhat different to what is typically used for higher-order QED calculations. For a start, infrared singularities due to soft photons are regularised through dimensional regularisation in $d = 4 - 2\epsilon$ dimensions. The photon is kept strictly massless also in intermediate steps. However, the masses of the fermions are always kept at their physical value and not set to zero. This regularises all collinear singularities in QED and gives rise to terms involving the logarithm of the fermion mass, $\log(m^2/Q^2)$. Such terms often form the dominant corrections in QED and, thus, it is essential to keep fermion masses different from zero. This leads to a substantial complication in the evaluation of virtual corrections. If the two-loop amplitudes are available only for massless fermions, massification [16–20] can be used. This is a procedure that allows to obtain the leading mass terms from the corresponding massless amplitudes. Two-loop amplitudes obtained this way are suitable for our approach, but result in the neglect of usually very small power suppressed terms. While it is possible to partially resum logarithmic terms, at the current stage no effort is made within MᴄMᴜʟᴇ to do so. Presently, MᴄMᴜʟᴇ is a strict fixed-order fully differential particle/parton-level Monte Carlo integrator.

Since we are dealing with low-energy processes we always renormalise the fermion masses and coupling in the on-shell scheme. The treatment of infrared singularities that occur when combining real and virtual corrections is coded according to the FKS subtraction method [21, 22] and its generalisation beyond NLO for massive QED, FKS$^{\ell}$ [13].

The core idea of this method is to render the phase-space integration of a real matrix element finite by subtracting all possible soft limits. The subtracted pieces are partially integrated over the phase space and combined with the virtual matrix elements to form finite integrands. For a detailed discussion of the method we refer to [13]. Here, we just give a schematic overview with the basic information required to understand the structure of the code.

The NLO corrections $\sigma^{(1)}$ to a cross section are split into a $n$-particle and $(n+1)$-particle contribution and are written as

$$\sigma^{(1)} = \sigma_n^{(1)}(\xi_c) + \sigma_{n+1}^{(1)}(\xi_c), \tag{2a}$$

$$\sigma_n^{(1)}(\xi_c) = \int \, \mathrm{d}\Phi_n^{d=4} \left( \mathcal{M}_n^{(1)} + \hat{\mathcal{E}}(\xi_c) \mathcal{M}_n^{(0)} \right) = \int \, \mathrm{d}\Phi_n^{d=4} \, \mathcal{M}_n^{(1)f}(\xi_c), \tag{2b}$$

$$\sigma_{n+1}^{(1)}(\xi_c) = \int \, \mathrm{d}\Phi_{n+1}^{d=4} \left( \frac{1}{\xi_1} \right)_c \left( \xi_1 \mathcal{M}_{n+1}^{(0)f} \right). \tag{2c}$$

In (2c), $\xi_1$ is a variable of the $(n+1)$-parton phase space $\mathrm{d}\Phi_{n+1}^{d=4}$ that corresponds to the (scaled) energy of the emitted photon. For $\xi_1 \to 0$ the real matrix element (or more precisely the absolute value squared of the amplitude) $\mathcal{M}_{n+1}^{(0)f}$ develops a singularity. The superscripts (0) and $f$ indicate that the matrix element is computed at tree level and is finite, i.e. free of explicit infrared poles $1/\epsilon$. In order to avoid an implicit infrared pole upon integration, the $\xi_1$ integration is modified by the factor $\xi_1(1/\xi_1)_c$, where the distribution $(1/\xi_1)_c$ acts on a test function $f(\xi_1)$ as

$$\int_0^1 \mathrm{d}\xi_1 \left( \frac{1}{\xi_1} \right)_c f(\xi_1) \equiv \int_0^1 \mathrm{d}\xi_1 \frac{f(\xi_1) - f(0)\theta(\xi_c - \xi_1)}{\xi_1}. \tag{3}$$

Thus, for $\xi_1 < \xi_c$, the integrand is modified through the subtraction of the soft limit $f(0)$. This renders the integration finite. However, it also modifies the result. The missing piece of the real corrections can be trivially integrated over $\xi_1$. This results in the integrated eikonal factor $\hat{\mathcal{E}}(\xi_c)$ times the tree-level matrix element for the $n$-particle process, $\mathcal{M}_n^{(0)}$. The factor $\hat{\mathcal{E}}(\xi_c)$ has

an explicit $1/\epsilon$ pole that cancels precisely the corresponding pole in the virtual matrix element $\mathcal{M}_n^{(1)}$. Thus, the combined integrand of (2b) is free of explicit poles, hence denoted by $\mathcal{M}_n^{(1)f}$, and can be integrated numerically over the $n$-particle phase space $d\Phi_n^{d=4}$.

The parameter $\xi_c$ that has been introduced to split the real corrections can be chosen arbitrarily as long as

$$0 < \xi_c \leq \xi_{\max} = 1 - \frac{\left(\sum_i m_i\right)^2}{s}, \tag{4}$$

where the sum is over all masses in the final state. The $\xi_c$ dependence has to cancel exactly between (2b) and (2c) since at no point any approximation was made in the integration. Checking this independence is a very useful tool to test the implementation of the method as well as its numerical stability.

The finite matrix element $\mathcal{M}_n^{(1)f}$ is simply the first-order expansion of the general YFS exponentiation formula [12] for soft singularities

$$e^{\hat{\mathcal{E}}} \sum_{\ell=0}^{\infty} \mathcal{M}_n^{(\ell)} = \sum_{\ell=0}^{\infty} \mathcal{M}_n^{(\ell)f} = \mathcal{M}_n^{(0)} + \left(\mathcal{M}_n^{(1)} + \hat{\mathcal{E}}(\xi_c)\mathcal{M}_n^{(0)}\right) + \mathcal{O}(\alpha^2), \tag{5}$$

where we exploited the implicit factor $\alpha$ in $\hat{\mathcal{E}}$.

As detailed in [13], for QED with massive fermions this scheme can be extended to NNLO and, in fact, beyond. The NNLO corrections are split into three parts

$$\sigma_n^{(2)}(\xi_c) = \int d\Phi_n^{d=4}\left(\mathcal{M}_n^{(2)} + \hat{\mathcal{E}}(\xi_c)\mathcal{M}_n^{(1)} + \frac{1}{2!}\mathcal{M}_n^{(0)}\hat{\mathcal{E}}(\xi_c)^2\right) = \int d\Phi_n^{d=4}\,\mathcal{M}_n^{(2)f}(\xi_c), \tag{6a}$$

$$\sigma_{n+1}^{(2)}(\xi_c) = \int d\Phi_{n+1}^{d=4}\left(\frac{1}{\xi_1}\right)_c \left(\xi_1 \mathcal{M}_{n+1}^{(1)f}(\xi_c)\right), \tag{6b}$$

$$\sigma_{n+2}^{(2)}(\xi_c) = \int d\Phi_{n+2}^{d=4}\left(\frac{1}{\xi_1}\right)_c \left(\frac{1}{\xi_2}\right)_c \left(\xi_1\xi_2 \mathcal{M}_{n+2}^{(0)f}\right). \tag{6c}$$

Thus we have to evaluate $n$-parton contributions, single-subtracted $(n+1)$-parton contributions, and double-subtracted $(n+2)$-parton contributions. This structure will be mirrored in the Fortran code. The $\xi_c$ dependence cancels, once all three contributions are taken into account. An example of this will be shown in Figure 6.

The method described above has actually already been used for several processes. The radiative [23] and rare decay [24] of the muon and tau [25] have been implemented at NLO in the Fermi theory in a fully differential code. In addition, the Michel decay of the muon has been added at NNLO [13]. These results have been verified by comparison to more analytic and more inclusive computations [26–30]. Thus, the method is fully established and McMule can be seen as a natural extension of these previous computations and a container to include further phenomenologically relevant processes.

## 3 Structure of McMule

McMule is written in Fortran 95 with helper and analysis tools written in `python`[1]. An online documentation can be found at the git repository listed in the introduction [31]. The code is written with two kinds of applications in mind. First, several processes are implemented, some at NLO, some at NNLO. Since new processes are continuously added, we refer to the

---

[1]Additionally to the `python` tool a Mathematica tool is available.

online documentation for a list of available processes. For these, the user can define an arbitrary (infrared safe), fully differential observable and compute cross sections and distributions. Second, the program is set up such that additional processes can be implemented by supplying the relevant matrix elements.

To obtain a copy of McMule we recommend the following approach

```
$ git clone --recursive https://gitlab.com/mule-tools/mcmule
```

To build McMule, a Fortran compiler such as `gfortran` and a python installation is needed. The main executable can be compiled by running

```
$ ./configure
$ make mcmule
```

Alternatively, we provide a Docker container [32] for easy deployment and legacy results. In multi-user environments, *udocker* [33] can be used instead. In either case, a pre-compiled copy of the code can be obtained by calling

```
$ docker pull yulrich/mcmule  #requires Docker to be installed
$ udocker pull yulrich/mcmule #requires uDocker to be installed
```

We provide instructions on how McMule is used in Section 4.

When started, `mcmule` reads options from `stdin` as specified in Table 1 of Section 4. The value and error estimate of the integration is printed to `stdout` and the full status of the integration is written in a machine-readable format into a folder called `out/` (see below).

McMule consists of several modules with a simple, mostly hierarchic structure. The relation between the most important Fortran modules is depicted in Figure 1. A solid arrow indicates "using" the full module, whereas a dashed arrow is indicative of partial use. In what follows we give a brief description of the various modules and mention some variables that play a prominent role in the interplay between the modules.

`global_def:` This module simply provides some parameters such as fermion masses that are needed throughout the code. It also defines `prec` as a generic type for the precision used.[2] Currently, this simply corresponds to double precision.

`functions:` This is a library of basic functions that are needed at various points in the code. This includes dot products, eikonal factors, the integrated eikonal, and an interface for scalar integral functions among others.

`collier:` This is an external module [34–37]. It will be linked to McMule during compilation and provides the numerical evaluations of the scalar and in some cases tensor integral functions in `functions`.

`phase_space:` The routines for generating phase-space points and their weights are collected in this module. Phase-space routines ending with FKS are prepared for the FKS subtraction procedure with a single unresolved photon. In the weight of such routines a factor $\xi_1$ is omitted to allow the implementation of the distributions in the FKS method, see (2c). This corresponds to a global variable `xiout`. This factor has to be included in the integrand of the module `integrands`. Also the variable `ksoft` is provided that corresponds to the photon momentum without the (vanishing) energy factor $\xi_1$. Routines ending with FKSS are routines with two unresolved photons, see (6c). Correspondingly, a factor $\xi_1 \xi_2$ is missing in the weight. The global variables `xiout1` and `xiout2` as well as `ksoft1` and `ksoft2` are provided.[3]

---

[2]For quad precision `prec=16` and the compiler flag `-fdefault-real-16` is required.

[3]In the current version of McMule these variables are called `xioutA`, `xioutB`, `ksoftA`, and `ksoftB`.

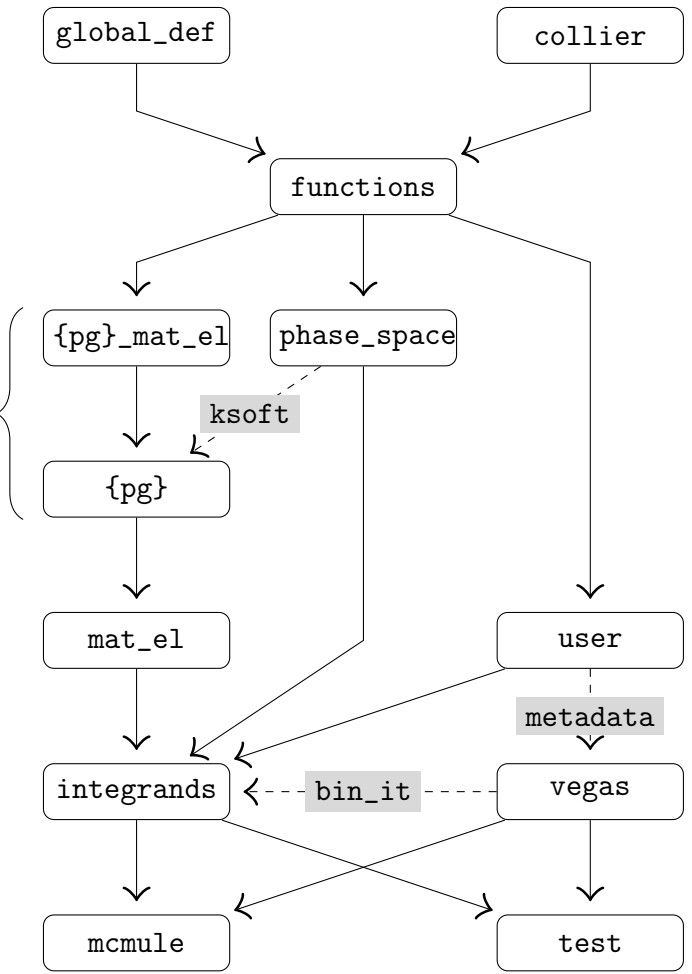

Figure 1: The structure of McMule

{pg}_mat_el: Matrix elements are grouped into process groups such as muon decay (`mudec`) or $\mu$-$e$ and $\mu$-$p$ scattering (`mue`). Each process group contains a `mat_el` module that provides all matrix elements for its group. Simple matrix elements are coded directly in this module. More complicated results are imported from sub-modules not shown in Figure 1. A matrix element starting with P contains a polarised initial state. A matrix element ending in `av` is averaged over a neutrino pair in the final state.

{pg}: In this module the soft limits of all applicable matrix elements of a process group are provided to allow for the soft subtractions required in the FKS scheme. These limits are simply the eikonal factor evaluated with `ksoft` from `phase_space` times the reduced matrix element, provided through `mat_el`.

   This module also functions as the interface of the process group, exposing all necessary functions that are imported by

mat_el, which collects all matrix elements as well as their particle labelling or particle identification.

user: For a user of the code who wants to run for an already implemented process, this is the only relevant module. At the beginning of the module, the user has to specify the number of quantities to be computed, `nr_q`, the number of bins in the histogram, `nr_bins`, as well as their lower and upper boundaries, `min_val` and `max_val`. The last three quantities are arrays of length `nr_q`. The quantities themselves, i.e. the measurement function,

is to be defined by the user in terms of the momenta of the particles in `quant`. Cuts can be applied by setting the logical variable `pass_cut` to false[4]. Some auxiliary functions like (pseudo)rapidity, transverse momentum etc. are predefined in `functions`. Each quantity has to be given a name through the array `names`.

Further, `user` contains a subroutine called `inituser`. This allows the user to read additional input at runtime, for example which of multiple cuts should be calculated. It also allows the user to print some information on the configuration implemented.

`vegas`: As the name suggests this module contains the adaptive Monte Carlo routine `vegas` [38]. The binning routine `bin_it` is also in this module, hence the need for the binning metadata, i.e. the number of bins and histograms (`nr_bins` and `nr_q`, respectively) as well as their bounds (`min_val` and `max_val`) and names, from `user`.

`integrands`: In this module the functions that are to be integrated by `vegas` are coded. There are three types of integrands: non-subtracted, single-subtracted, and double-subtracted integrands, corresponding to the three parts of (6). The matrix elements to be evaluated and the phase-space routines used are set using function pointers through a subroutine `initpiece`. The factors $\xi_i$ that were omitted in the phase-space weight have to be included here for the single- and double-subtracted integrands.

`mcmule`: This is the main program, but actually does little else than read the inputs and call `vegas` with a function provided by `integrands`.

`test`: For developing purposes, a separate main program exists that is used to validate the code after each change. Reference values for matrix elements and results of short integrations are stored here and compared against.

The library of matrix elements deserves a few comments. As matrix elements quickly become very large, we store them separately from the main code. This makes it also easy to extend the program by minimising the code that needs to be changed.

We group the various contributions into *process group*, *generic processes*, and *generic pieces* as indicated in Figure 2. The generic process is a prototype for the physical process such as $\ell p \to \ell p$ (cf. Section 6) where the flavour of the lepton $\ell$ is left open. The generic piece describes a part of the calculation such as the real or virtual corrections, i.e. the different pieces of (2) (or correspondingly (6) at NNLO), that themselves may be further subdivided as is convenient. In particular, in some cases a generic piece is split into various phase-space partitions, as in the example of `em2emREE` in Figure 2. A more detailed listing of the various contributions required for $\mu$-$e$ scattering is given in Figure 5.

When running `mcmule`, the code generates a statefile from which the full state of the integrator can be reconstructed should the integration be interrupted. This makes the statefile ideal to also store results in a compact format. To analyse these results, we provide a python tool `pymule`, additionally to the main code for MCMULE. The tool `pymule`, which can be found under `tools/pymule`, uses `numpy` [39] for data storage and `matplotlib` for plotting [40]. While `pymule` works with any python interpreter, `IPython` [41] is recommended. A full list of functions provided can be found in the online manual of `pymule` [31].

An important issue are numerical instabilities arising in problematic regions of the phase space. This is typically the case if an emitted photon becomes soft or collinear to a massive, but light, fermion. For soft photon emission the numerical instability is related to the FKS subtraction discussed in Section 2. When $\xi_1$ becomes very small, the difference $f(\xi_1) - f(0)\theta(\xi_c - \xi_1)$ in (3) becomes potentially troublesome as $f(\xi_1)$ can be calculated less precisely than $f(0)$. To

---

[4]Technically, `pass_cut` is a list of length `nr_q`, allowing to decide whether to cut for each histogram separately.

McMule
- process group `mudec`
  - generic process `m2enn`: $\mu \to \nu\bar{\nu}e$, $\tau \to \nu\bar{\nu}e$, $\tau \to \nu\bar{\nu}\mu$
    - generic piece `m2enn0`
    - generic piece `m2ennF`
    - generic piece `m2ennR`
  - generic process `m2enng` $\mu \to \nu\bar{\nu}e\gamma$, $\tau \to \nu\bar{\nu}e\gamma$, $\tau \to \nu\bar{\nu}\mu\gamma$
    - generic piece `m2enng0`
    - generic piece `m2enngV`
    - generic piece `m2enngC`
    - generic piece `m2enngR`
  - generic process `m2ennee`: $\mu \to \nu\bar{\nu}eee$, $\tau \to \nu\bar{\nu}eee$, $\tau \to \nu\bar{\nu}e\mu\mu$, $\tau \to \nu\bar{\nu}\mu\mu\mu$,
    ...
    - generic piece `m2ennee0`
    - ...
- process group `mue`
  - generic process `em2em`: $\mu e \to \mu e$
    - generic piece `em2em0`
    - generic piece `em2emFEE`
    - generic piece `em2emREE`
      - partition `em2emREE15`
      - partition `em2emREE35`
    - ...
  - generic process `mp2mp`: $\mu p \to \mu p$, $ep \to ep$
    - generic piece `mp2mp0`
    - generic piece `mp2mpF`
    - ...
- ...

Figure 2: The structure of process group, generic process, and generic piece as used by McMule. The suffices 0, V, C, F, R, and others are explained in more detail in Section 5.

avoid this, we choose a very small `softcut`, below which we set the integrand directly to zero. In the collinear case small fermion masses give rise to pseudo-collinear singularities that further complicate a numerical stable evaluation of the matrix element. McMule addresses this issue through a dedicated tuning of the phase-space parametrisation to help the `vegas` integration find and deal with these problematic regions. In addition, a `collcut` is applied if the photon becomes very collinear to a light fermion. During development, `softcut` and `collcut` are varied to make sure that, within the integration error, the cross section is independent of the chosen values. Afterwards, a suitable value is chosen and hard-coded. However, the user retains the ability to modify this in `inituser`.

# 4 Running McMule: double radiative muon decay as an example

In order to provide a simple example with concrete instructions on how to run the code and to illustrate how it works, we consider the double radiative decay of the muon $\mu \to e[\nu\bar{\nu}]\gamma\gamma$ at leading order. Since the neutrinos are not detected, we average over them, indicated by the brackets. Hence, we have to be fully inclusive with respect to the neutrinos. But the code allows to make any cut on the other final-state particles.

To be concrete let us assume we want to compute two distributions, the missing energy $\not{E} \equiv E(\mu) - E(e) - E(\gamma_1) - E(\gamma_2)$ and $\cos\theta_e$, the cosine of the angle between the outgoing positron and the muon polarisation. Both quantities are determined in the rest frame of the decaying muon. Of course, $\not{E}$ corresponds to the combined energies of the neutrinos. To avoid an infrared singularity in the branching ratio, we have to require a minimum energy of the photons. We choose this to be $E_\gamma \geq 10$ MeV individually for both photons. In addition, we require for the angle between the two photons $\theta_{\gamma\gamma} > 15°$.

As mentioned in Section 3 the quantities are defined in the module user (file `src/user.f95`). At the beginning of the module we set

```
nr_q = 2
nr_bins = 50
min_val = (/ 0._prec , -1._prec /)
min_val = (/ 50._prec , 1._prec /)
```

where we have decided to have 50 bins for both distributions and `nr_q` determines the number of distributions. The boundaries for the distributions are set as $0 < \not{E} < 50$ MeV and $-1 \leq \cos\theta_e \leq 1$.

The quantities themselves are defined in the function `quant`. This function takes arguments q1 to q7. These are the momenta of the particles, arrays of length 4 with the fourth entry the energy. To figure out which momentum corresponds to which particle the user needs to check the headers in the module `mat_el` or in the manual [31]. In our case, we find

```
!! From file mudec_pm2ennggav.f95
use mudec , only: pm2ennggav!!(p1, n1, p2, p3, p4, p5, p6)
!! mu+(p1) -> e+(p2) \nu_e \bar{\nu}_\mu g(p5) g(p6)
!! mu-(p1) -> e-(p2) \bar{nu}_e  \nu_\mu g(p5) g(p6)
!! for massive (and massless) electron
!! average over neutrino tensor taken
```

Indicating that we have p1 for the incoming $\mu$, p2 for the outgoing $e$, and p5 and p6 for the two outgoing photons. The momenta of the neutrinos must be given but do not enter, as we average over them. Schematically, the function `quant` might look like

Table 1: The options read from `stdin` by McMule. The calls are multiplied by 1000.

| Variable name | Data type | Comment |
|---|---|---|
| nenter_ad | integer | calls / iteration during pre-conditioning |
| itmx_ad | integer | iterations during pre-conditioning |
| nenter | integer | calls / iteration during main run |
| itmx | integer | iterations during main run |
| ran_seed | integer | random seed |
| xinormcut1 | real(prec) | the $0 < \xi_c \leq 1$ parameter |
| xinormcut2 | real(prec) | the second $\xi_c$ parameter for NNLO (or the $\delta_{\text{cut}}$) |
| which_piece | char(20) | the part of the calculation to perform |
| flavour | char(8) | the particles involved |
| (opt) | unknown | the user can request further input during `userinit` |

```
FUNCTION QUANT(P1,P2,P3,P4,P5,P6,P7)
.
.
pass_cut = .true.
pol1 = (/ 0._prec, 0._prec, 0.85_prec, 0._prec /)

ez = (/ 0._prec, 0._prec, 1._prec, 0._prec /)

if(p5(4) < 10._prec .or. p6(4) < 10._prec) pass_cut = .false.
if(cos_th(p5,p6) > 0.965926) pass_cut = .false.

Emiss = p1(4)-p2(4)-p5(4)-p6(4)
names(1) = 'Emiss'
quant(1) = emiss
names(2) = 'CangE'
quant(2) = cos_th(p2,ez)

END FUNCTION QUANT
```

Here, we have used the function `cos_th` provided by the module `functions`. This returns the cosine of the angle between the two momenta given as arguments. We have also specified the polarisation vector `pol1` in accordance with the $\mu^+$ beam used by MEG. This polarisation has been measured [42] to be $P_\mu = -0.85 \pm 0.05$. Since McMule defines the polarisation through $\mu^-$, the sign has to be changed. The variable `pass_cut` controls the cuts. Initially it is set to true, to indicate that the event is kept. Applying a cut amounts to setting `pass_cut` to false.

All that remains to be done is to prepare the input read by `mcmule` from `stdin`, as specified in Table 1.

To be concrete let us assume we want to use 10 iterations with $1000 \times 10^3$ points each for pre-conditioning and 20 iterations with $5000 \times 10^3$ points each for the actual numerical evaluation. We pick a random seed, say 24225, and for the input variable `which_piece` we enter m2enngR. Since the double radiative muon decay is not on the list of processes (1), we actually compute the real corrections (hence the suffix R) of the generic process $\mu \to e \nu \bar\nu \gamma$. The `flavour` variable is set to mu-e. We could e.g. use tau-e to change from the generic process $\mu \to e \nu \bar\nu \gamma$ to the process $\tau \to e \nu \bar\nu \gamma$. This system will be used for other processes as well. The input variable `which_piece` determines the generic process and the part of it that is to be computed (i.e. tree level, real, double-virtual etc.). In a second step, the input `flavour` associates actual numbers to the parameters entering the matrix elements and phase-space generation.

Thus, we run the code by giving the input

```
$ ./mcmule
1000
10
5000
20
24225
0.1
0.1
m2ennR
mu-e
```

In practice the input will typically not be given by hand. We mention a more efficient way in Section 5 as well as the manual [31]. The two variables `xinormcut1` and `xinormcut2` have no effect at all for a tree-level calculation and will be discussed in Section 5.1 in the context

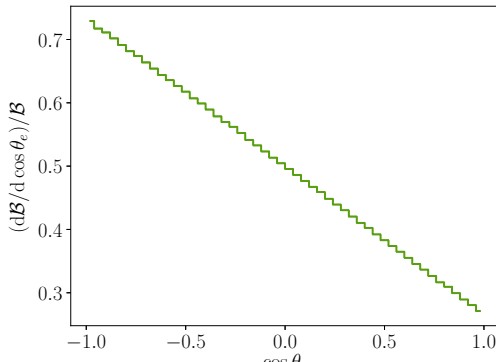
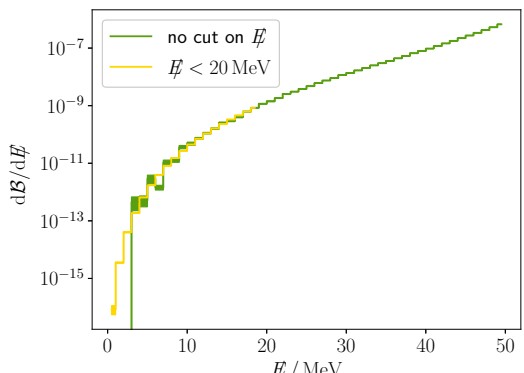

Figure 3: Results of a short test run for the branching ratio at LO for the double radiative muon decay $\mu \to e\nu\bar{\nu}\gamma\gamma$, as a function of the missing energy and the angle of the outgoing positron. For the region $\not{E} < 20\,\text{MeV}$ a tailored run is shown in yellow.

of the NLO and NNLO run for muon-electron scattering. We also ignore the optional input for the moment.

Alternatively, McMULE can be run using Docker or *udocker* without compiling it first by running

```
$ ./tools/run-docker.sh -i yulrich/mcmule:latest \
-u path/to/user.f95  -r
```

followed by the same input as above.

Now the mule is ready to trot. After about fifteen minutes on an Intel i5 processor, it returns the total cross section as[5]

```
result, error:  {  1.10052E+06,  1.80838E+02 };   chisq:  0.83
```

which, after adding the factor $G_F^2\alpha^2/\Gamma_\mu$ results in a branching ratio of $\mathcal{B} = 2.6611(4) \cdot 10^{-5}$. Here $\Gamma_\mu$ is the measured width of the muon and $G_F$ the Fermi constant as given in the Appendix A. The error is only the statistical error of the Monte Carlo and not a theory error. In addition, the two distributions are written into a binary file that contains the full state of the integrator `out/m2enngR_mu-e_S0000024225X1.00000D1.00000_ITMX020x005M.vegas`. The corresponding results are shown as green histograms in Figure 3, where $d\mathcal{B}/d\cos\theta_e$ has been normalised.

The results have rather poor statistics. In particular the precision in the low-energy tail of the sharply falling $\not{E}$ distribution is very low since the Monte Carlo generates very few points there. If the user is interested in this tail it is advisable to perform dedicated runs. This can be done simply by adding a cut like

```
if(emiss > 20.) pass_cut = .false.
```

in `quant`. The Monte Carlo will then adapt and result in a more precise determination of the $\not{E}$ distribution in the region $\not{E} < 20\,\text{MeV}$. For illustration in Figure 3 such a tailored run with the same statistics is overlaid in yellow to the original run in the plot for $\not{E}$.

This is all that is required for simply running the code. In what follows we give a brief outline how the code works. The first step it does in `mcmule` is to associate the numerical values of the masses, as specified through `flavour`. In particular, the generic masses `Mm` and

---

[5]Unless otherwise stated, all numerical results have been obtained by running the code in Docker or *udocker*.

Me are set to `Mmu` and `Mel`. This is done in `initflavour(scms)`, defined in `global_def`. For other processes this might also involve setting e.g. centre-of-mass energies `scms` to default values.

Next, the function to be integrated by `vegas` is determined. This is a function stored in `integrands`. There are basically three types of integrands: a standard, non-subtracted integrand `sigma_0`, a single-subtracted integrand `sigma_1` needed beyond LO, and a double-subtracted integrand `sigma_2` needed beyond NLO. It is the variable `which_piece` that determines which of the three functions is called. Usually, for a LO case, we only need `sigma_0`. However, since the process $\mu \to e\nu\bar{\nu}\gamma\gamma$ as such is not implemented in MCMULE, we compute it at LO by calling the real corrections of the radiative muon decay $\mu \to e\nu\bar{\nu}\gamma$. Thus, from a technical point of view we call a single-subtracted integrand. The function `quant`, however, is constructed such that no subtraction takes place. This is ensured by the demand $E_\gamma > 10\,\mathrm{MeV}$. In addition, `which_piece` determines `ndim`, the dimension of the integration (11 in our case), and the matrix element that needs to be called, `Pm2ennggAV(q1,n1,q2...q6)`. The name of the function suggests we compute $\mu(q_1, n_1) \to e(q_2)[\nu\bar{\nu}]\gamma(q_5)\gamma(q_6)$ with the polarisation vector `n1` of the initial lepton, and the neutrinos are averaged over. Note that the momenta of the neutrinos are given as arguments, even if they are redundant. This simplifies the code a lot because it means that all matrix elements have the same calling convention.

The interplay between the function `sigma_1(x,wgt,ndim)` and `vegas` is as usual, through an array of random numbers `x` of length `ndim`. In addition the vegas weight of the event, `wgt`, is passed. The function `sigma_1` simply evaluates the complete weight `wg` of a particular event by combining `wgt` with the matrix element supplemented by symmetry, flux, and phase-space factors. In a first step a phase-space routine of `phase_space` is called. For our calculation this is the optimised phase space `psd6_p_25_26_m50_fks(x,p1,Mm,p2,Me,p3...p6,0.,` `weight)` generating the momenta with correct masses as well as the phase-space weight `weight`. The `d` in the name of the phase-space routine indicates that we are considering a decay process (one initial state particle), the `6` indicates the total number of momenta generated and the meaning of `fks` will be explained below. The other labels indicate the particular tuning and partition which are irrelevant in this case. With these momenta the observables to be computed are evaluated with a call to `quant`. If one of them passes the cuts, the variable `cuts` is set to true. This triggers the computation of the matrix element and the assembly of the full weight. In a last step, the routine `bin_it`, stored in `vegas`, is called to put the weight into the correct bins of the various distributions. These steps are done for all events and those after pre-conditioning are used to obtain the final distributions.

Since, technically speaking, we are computing a subtracted matrix element, the code also generates for each event the associated soft event, i.e. the same event with $\xi_1 \to 0$. This is realised by having a parametrisation of the phase space, such that setting the first entry of `x` to 0 results in $\xi_1 \to 0$. Such a phase-space routine is called FKS compatible and named with the ending `fks`. It is then checked whether the subtraction condition $\xi_1 < \xi_c$, (3), is satisfied. If yes, `quant` is evaluated with this new set of momenta, and if the event passes, the soft limit of the matrix element is evaluated and the subtraction is performed according to (3). The global variable `xiout` is required for this, since it is left out of the FKS phase-space weight and has to be included in the integrand. In our case, the soft event never passes the cuts, due to the requirement `q6(4) > 10.` in `quant`.

To conclude this section, we mention that the process considered here is actually relevant to searches for lepton-flavour violating decays mediated by a light particle $X$. Indeed, double radiative muon decay $\mu \to e\nu\bar{\nu}\gamma\gamma$ in the region of very small $\not{E}$ cannot be distinguished from $\mu \to eX$ with $X \to \gamma\gamma$.

MEG has performed a search for this decay [43]. In order to assess the background from

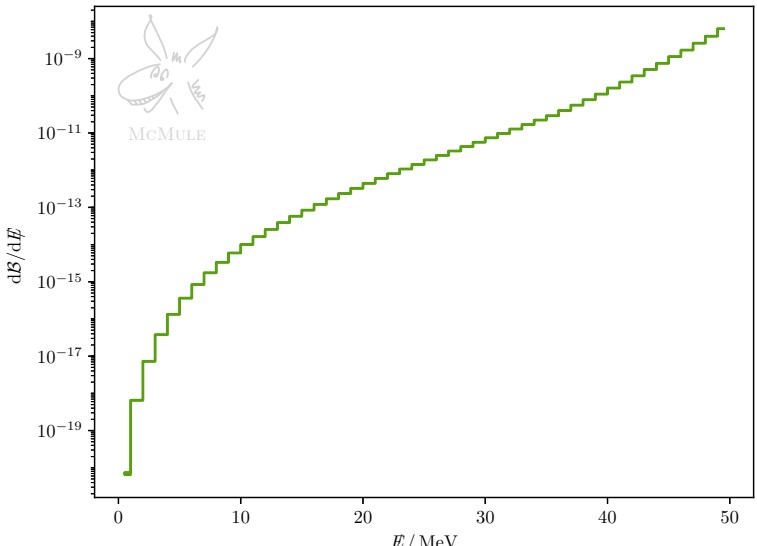

Figure 4: Branching ratio at LO for the double radiative muon decay $\mu \to e\nu\bar{\nu}\gamma\gamma$, as a function of the missing energy, i.e. the energy of the neutrinos.

double radiative muon decay, we have computed the $\not{E}$ distribution, with cuts

$$E_\gamma > 10\,\text{MeV}, \qquad |\cos\theta_\gamma| < 0.35\,, \qquad |\phi_\gamma| > \frac{2\pi}{3}\,, \qquad (7)$$

adapted to the MEG detector. Here, $\theta_\gamma$ and $\phi_\gamma$ are the polar and azimuthal angles of the two photons. Further, we require that the two photons can be separated in the calorimeter. This is implemented by specifying them to be $\delta x = 20\,\text{cm}$ apart on the detector surface which is at a radius of $R = 67.85\,\text{cm}$ resulting in

$$\theta_{\gamma\gamma} > \tan^{-1}\left(\frac{\delta x}{R}\right) \approx 16.4°\,. \qquad (8)$$

The results are shown in Figure 4, where special emphasis has been given to the small $\not{E}$ region. Integrating the differential distribution up to $\not{E} = 10\,\text{MeV}$ yields a branching ratio of $\mathcal{B}(\not{E} < 10\,\text{MeV}) = 1.2 \times 10^{-14}$.

## 5 Muon-electron scattering with McMule

Muon-electron scattering is a classic process which at low energy is completely dominated by QED. There is renewed interest in this process in connection with the long-standing $(3-4)\sigma$ discrepancy between the anomalous magnetic moment or $(g-2)_\mu$ of the muon and its Standard Model prediction. The theory calculation of $(g-2)_\mu$ suffers from uncertainties originating from non-perturbative hadronic corrections. The largest source of this uncertainty is the HVP, followed by the contribution due to hadronic light-by-light scattering [44]. A better understanding of the hadronic contributions is therefore of utmost importance, even more so in light of the new $(g-2)_\mu$ experiments at Fermilab [45] and J-Parc [46] that will further increase the experimental precision achieved by the BNL E831 experiment [47].

The HVP correction can be related to measurement data of electron-positron annihilation using a dispersive approach [48,49]. The resulting integrand is, however, highly fluctuating due to hadronic resonances and threshold effects. This makes the corresponding analysis rather

challenging. Furthermore, a recent lattice evaluation [50] of the HVP contribution to $(g-2)_\mu$ substantially deviates from the dispersive approach.

In this context it has been proposed to extract the HVP corrections from the measurement of the running of the QED coupling in the space-like region [51]. Contrary to the traditional time-like approach, the corresponding integrand is smooth and free of resonances. Moreover, this would yield an independent determination of the HVP contribution resulting, in turn, in a better understanding of the theory error.

While the original proposal was based on Bhabha scattering [51–53], it was recently established that the elastic scattering of muons on atomic electrons could, in principle, be sufficiently sensitive to reach a competitive precision with this novel approach [54]. This is the objective of the proposed MUonE experiment [8]. Since in this case the effect of the HVP to the running of the QED coupling for muon-electron scattering ranges from $10^{-3}$ to $10^{-5}$, the differential cross section would have to be measured at a precision of 10 ppm.

The radiative corrections to muon-electron scattering represent one source of systematic uncertainty that has to be carefully studied [55]. To reach the target precision these corrections have to be known at a level of 10 ppm as well. The effect of hadronic corrections was recently addressed in [56,57] using two independent methods. At leading order also $Z$-exchange has to be taken into account. The main corrections are, however, due to QED radiation. The minimal requirement is expected to be the NNLO QED corrections, for which we have to consider up to two photons in the final state

$$e^-(p_1)\mu^-(p_2) \rightarrow e^-(p_3)\mu^-(p_4)\left\{\gamma(p_5)\gamma(p_6)\right\}, \tag{9}$$

matched to leading-logarithmic resummation. In this section we report on the progress towards this goal made through MᴄMᴜʟᴇ.

## 5.1 Running MᴄMᴜʟᴇ for muon-electron scattering

The NLO QED corrections to muon-electron scattering have been known for a long time [58, 59]. Motivated by the MUonE experiment, they have been revisited and, together with the NLO electroweak corrections, implemented in a fully differential Monte Carlo code [15].

As a first step towards a sufficiently precise description of muon-electron scattering within MᴄMᴜʟᴇ, we have also implemented the NLO QED corrections. We have compared our results with [15] and found full agreement. MᴄMᴜʟᴇ also contains the dominant electronic NNLO corrections that are proportional to $Q_\mu^2 Q_e^6$, where $Q_\mu$ and $Q_e$ denote the charge of the muon and electron, respectively [55]. Also this part of the code is fully verified after comparing the observables defined in 'Setup 2' and 'Setup 4' of [15] with [60]. Since [15,60] use a photon mass as infrared regulator and the phase-space slicing method, the agreement is a strong cross check for a correct technical implementation. Details of the computation and physical results will be presented in Section 5.2. In this section we focus on a description on how to run the code.

There are several changes with respect to the example discussed in Section 4. First of all, the process is different. The generic process now is em2em. For a tree-level computation we can proceed analogous to Section 4 with `which_piece` set to em2em0. For a NLO computation we need to evaluate the virtual and real corrections. As shown in (2), using FKS this results in two terms, the subtracted real corrections (2c) and the finite virtual corrections (2b), i.e. the virtual corrections combined with the infrared counterterm. The corresponding `which_piece` are em2emR and em2emF, respectively.

The results obtained with em2emR and em2emF taken separately are $\xi_c$ dependent. This dependence has to cancel in the sum. The $\xi_c$ parameter is set through the variable `xinormcut1` of Table 1. The latter has to be set to a value between 0 and 1 and is related to $\xi_c$ through (4)

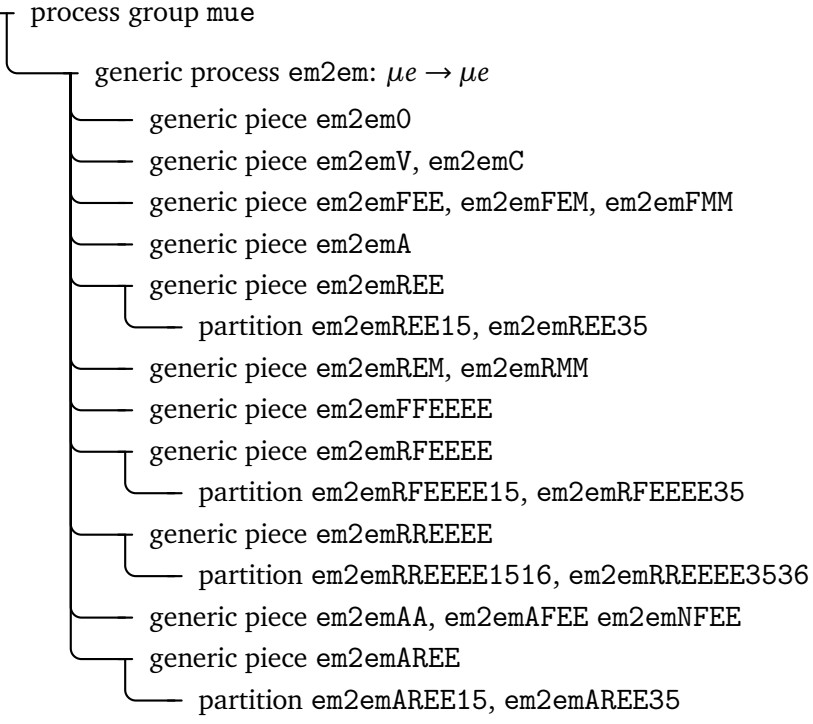

Figure 5: A complete list of contributions (`which_piece`) currently implemented for
$\mu$-$e$ scattering. This is a subset of Figure 2.

as `xinormcut1=` $\xi_c/\xi_{\max}$. Checking the independence of physical results on $\xi_c$ serves as a consistency check and is an implicit check on the infrared safety of the observable implemented in `quant`. To do this, it helps to disentangle `em2emF` into `em2emV` and `em2emC`, according to (2b).[6] The former corresponds to the pure virtual corrections whereas the latter is the infrared counterterm, i.e. the integrated eikonal times the tree-level matrix element. Of course, taken separately these terms are infrared divergent. MᴄMᴜʟᴇ returns the finite part, as defined in [31,61]. Only `em2emC` depends on $\xi_c$ and this part is typically much faster in the numerical evaluation.

In fact, `em2emR` and `em2emF` are divided up further, as can be seen in Figure 5, where a complete tree of possible `which_piece` for the generic process `em2em` is depicted. This additional separation corresponds to a gauge-invariant split of the NLO corrections into emission/absorption from the electron line EE, emission/absorption from the muon line MM, and the interference EM. As shown in [15], the EE contributions are by far dominant. In addition, these contributions suffer most from pseudo singularities that arise from photon emission nearly collinear to the electron. To deal with these regions of phase space in a numerically stable way, there is one further purely technical partitioning of `em2emREE` into `em2emREE15` and `em2emREE35`. These two partitions have a tuned phase space in $s_{15} = 2p_1 \cdot p_5$ and $s_{35} = 2p_3 \cdot p_5$, respectively, to deal with initial-state and final-state pseudo-collinear singularities.

Finally, we note that also hadronic contributions are implemented. This is done together with the leptonic vacuum polarisation (VP) in `em2emA`. The user can then set the variables `nel`, `nmu`, `ntau`, and `nhad` to decide which contributions to include. For the calculation of the HVP the Fortran library `alphaQED` [62–64] is used. Specifically, we rely on the hadronic stand-alone version `hadr5n12.f`.

Choosing different random seeds, varying $\xi_c$ and having to compute the various real and

---

[6]This additional split is not implemented for all processes.

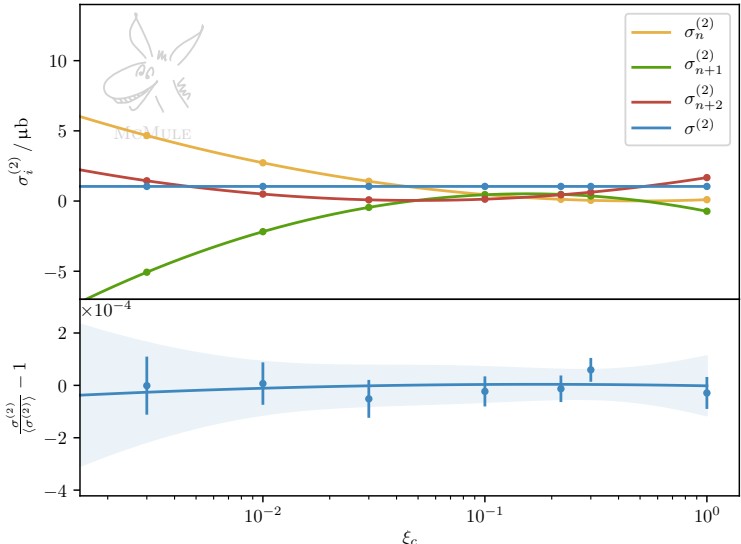

Figure 6: The (in)dependence of the pure NNLO contribution $\sigma^{(2)}$ to the total cross section on the arbitrary cut parameter $\xi_c$. The error band represents the $1\sigma$ confidence level of the fit.

virtual parts results in quite a few jobs. A particularly convenient way to run McMule is using menu files. A menu file contains a list of jobs to be computed such that the user will only have to vary the random seed and $\xi_c$ as the statistical requirements are defined globally in a config file. This is completed by a submission script, usually called submit.sh. The submit script is what will need to be launched. It will take care of the starting of different jobs. It can be run on a normal computer or on a Slurm cluster [65].

To prepare the run in this way we can use pymule, a tool provided together with McMule. When using pymule create, we are asked various questions, most of which have a default answer in square brackets. In the end pymule will create a directory, where all results will be stored. In addition pymule also provides tools to analyse the results, such as combining runs with different random seeds and different choices of $\xi_c$. A more detailed description of pymule can be found in the online documentation [31,61].

Moving from NLO to NNLO increases the number of partial results further. Now we have to run with which_piece set to em2emRREEEE (double-real corrections), em2emRFEEEE (real virtual corrections), and em2emFFEEEE (double-virtual corrections). As discussed above, the additional ending EEEE indicates that only electronic corrections are included. As for the real corrections, also the real-virtual and the double-real contributions are computed with a partition to disentangle initial-state and final-state pseudo-collinear singularities.

Also at NNLO the correction due to hadronic and leptonic VP is included. These contributions are split up according to the classification of [57]. The diagrams where the VP factorises are implemented in em2emAA, em2emAR, and em2emAF. The former takes into account diagrams with one or two insertions of the VP into the tree level diagram. The latter two implement QED NLO corrections combined with one insertion. The remaining non-factorisable vertex correction can be computed via em2emNF. This relies on the results of [56] which uses the hyperspherical integration method to calculate the hadronic corrections to muon-electron scattering [66].

As listed in Table 1, running at NNLO there are two $\xi_c$ variables to be set in the input.

However, to obtain $\xi_c$ independent physical results it is imperative that they are set equal, `xinormcut1=xinormcut2`. The reason MᴄMᴜʟᴇ works with two variables is that for computations with massless fermions, `xinormcut2` corresponds to the unphysical cut variable related to the collinear subtraction, often denoted by $\delta_{\text{cut}}$. Also, an independent `xinormcut2` can be used for internal checks.

An example of a typical check of the $\xi_c$ (in)dependence is shown in Figure 6, where the $n$-particle (orange), $(n+1)$-particle (green), and $(n+2)$-particle (red) contributions are shown separately for the total cross section according to 'Setup 4' of [15].[7] These are just the parts given in (6). In the sum (blue) the $\xi_c$ dependence cancels. This can be seen particularly well in the bottom panel, where the results of seven separate choices of $\xi_c$ are shown, together with a $1\sigma$ band of a fit to the $\xi_c$ dependence.

Once more we stress that this cancellation is exact. Thus, in principle any choice is allowed. However, for very small choices of $\xi_c$ there are large numerical cancellations. In the case of production runs, it is thus advisable to pick a value of $\xi_c$ where the separate contributions have roughly the same magnitude as the final result. From experience, a choice around $\xi_c \sim 0.1$ is a good starting point.

## 5.2 The dominant NNLO corrections

We now turn to the technical details of the calculation as well as the phenomenological discussion of the results. As previously mentioned, at NNLO we restrict ourselves to the gauge-invariant subgroup that only contains electronic corrections, i.e. contributions proportional to $Q_\mu^2 Q_e^6$. These corrections are expected to be dominant compared to the other contributions at this perturbative order as a consequence of enhanced collinear logarithms. To be consistent, at NNLO we therefore also only include VP with electrons inside the loop.

Furthermore, we assume the electron to be unpolarised in correspondence with the atomic electrons of the MUonE experiment. Our results are therefore independent of the muon polarisation due to parity invariance of QED. Additionally, the considered gauge-invariant subset is also independent of whether the muon beam consists of $\mu^+$ or $\mu^-$. This does not hold, however, for the full set of NLO QED corrections that is included here. This includes the muon and tau VP.

The double-virtual diagrams were calculated with the full electron mass dependence using the analytic expressions for the heavy quark form factors of [67]. Furthermore, the genuine two-loop corrections to the photon self-energy were taken from [68]. The diagrams for the real-virtual and double-real contributions were generated using QGraf [69] and calculated with Package-X [70]. An independent calculation was performed using FORM [71]. Complicated scalar triangle- and box-functions were then evaluated with the `COLLIER` library [34]. Additionally, `COLLIER` was used to perform a numerical stable tensor reduction in problematic regions of the phase space.

With the momenta of the particles labelled as in (9) we define the invariants $t_e = (p_1 - p_3)^2$ and $t_\mu = (p_2 - p_4)^2$. In the case of purely virtual corrections we have $t_e = t_\mu$. The energy of the outgoing electron and muon are denoted by $E_e$ and $E_\mu$, respectively. Additionally, we use $\theta_e$ and $\theta_\mu$ as the corresponding scattering angles relative to the beam axis. We further assume a muon beam of energy $E = 150\,\text{GeV}$, consistent with the M2 beam line at CERN North Area [8].

The total cross section is ill-defined due to the behaviour $d\sigma/dt \sim t^{-2}$ with $t_{\text{min}} \leq t \leq 0$. We therefore have to apply a cut on the maximal value of $t$ or equivalently on the minimal energy of the outgoing electron. In all of the results below we have chosen $E_e > 1\,\text{GeV}$. To model the geometry of the detector we require in addition that $\theta_\mu > 0.3\,\text{mrad}$.

---

[7]In fact, this was one of the numbers compared with [60].

Table 2: Results for the integrated cross section for S1 (without elasticity cut) and S2 (with elasticity cut) at LO, NLO, and NNLO. All digits given are significant compared to the error of the numerical integration.

|  | $\sigma/\mu b$ | | $\delta K^{(i)}/\%$ | |
|  | S1 | S2 | S1 | S2 |
|---|---|---|---|---|
| $\sigma^{(0)}$ | 121.4229 | 121.4229 | | |
| $\sigma^{(1)}$ | 0.5440 | −4.0773 | 0.4480 | −3.3580 |
| $\sigma^{(2)}$ | −0.0058 | +0.0093 | −0.0048 | 0.0079 |
| $\sigma_2$ | 121.9611 | 117.3549 | | |

Following [55], the outgoing electron and muon angles are in the absence of photons related through the elasticity condition

$$\tan\theta_\mu^{\text{el}} = \frac{2\tan\theta_e}{(1+\gamma^2\tan^2\theta_e)(1+g_\mu^*)-2}\,, \tag{10}$$

where

$$g_\mu^* = \frac{Em+M^2}{Em+m^2}\,, \qquad \gamma = \frac{E+m}{\sqrt{s}}\,, \tag{11}$$

with $s$ the centre-of-mass energy. This allows to restrict radiation with the elasticity cut

$$0.9 < \frac{\theta_\mu}{\theta_\mu^{\text{el}}} < 1.1\,. \tag{12}$$

In the following we present results with and without this additional cut, in order to analyse its impact on the radiative corrections. A similar effect can be expected as in the case of the acoplanarity cut of [15], where the NLO corrections flatten out significantly.

In summary, we consider the two scenarios

- S1: $E_e > 1\,\text{GeV}$, $\theta_\mu > 0.3\,\text{mrad}$,

- S2: $E_e > 1\,\text{GeV}$, $\theta_\mu > 0.3\,\text{mrad}$, $0.9 < \theta_\mu/\theta_\mu^{\text{el}} < 1.1$.

The order-by-order contributions, $\sigma^{(i)}$, to the integrated cross section, $\sigma_2 = \sigma^{(0)}+\sigma^{(1)}+\sigma^{(2)}$, are presented in Table 2.[8] It also shows the corresponding $K$ factors defined as

$$K^{(i)} = 1 + \delta K^{(i)} = \frac{\sigma_i}{\sigma_{i-1}}\,. \tag{13}$$

Figure 7 and Figure 8 then show differential results that are of interest to the MUonE experiment. In particular, we present distributions with respect to $\theta_e$ and $t_\mu$. The differential cross section at LO as well as at NNLO are displayed in the upper panels. In addition, the lower panels show the differential $K$ factors

$$\delta K^{(i)} = \frac{d\sigma^{(i)}/dx}{d\sigma_{i-1}/dx}\,, \tag{14}$$

with $x \in \{\theta_e, t_\mu\}$. In dotted lines, the $K$ factors without the inclusion of the VP are shown.

We first remark that the numerical error for the distribution $d\sigma/d\theta_e$ (Figure 7) is much smaller than for $d\sigma/dt_\mu$ (Figure 8). This is due to the fact that the cross section in the latter case is practically zero in most parts of the kinematically allowed region. As exemplified in Section 4, the statistics for $d\sigma/dt_\mu$ could be drastically improved using tailored runs. Nevertheless, the discontinuities of Figure 8 indicate that the Monte Carlo error for individual bins provided by McMule might be underestimated.

---

[8]For this paper, $\sigma^{(2)}$ only denotes the dominant NNLO contribution as defined in the various sections.

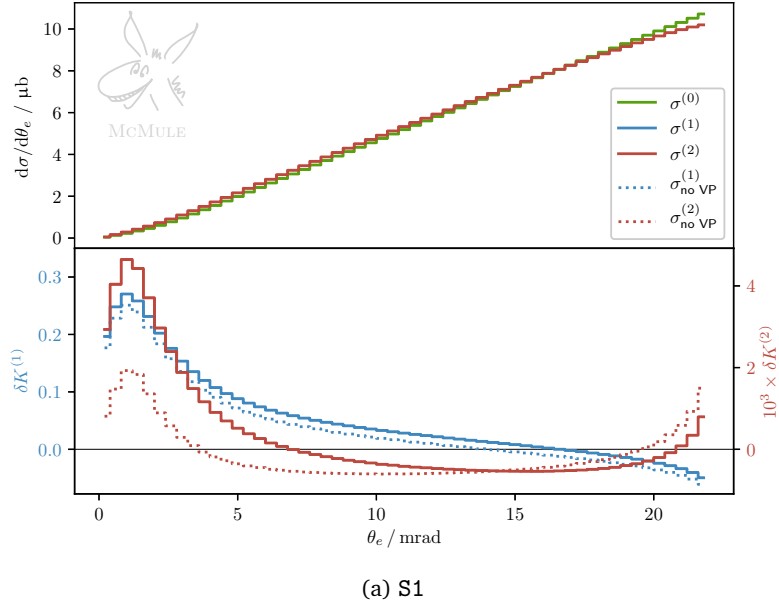

(a) S1

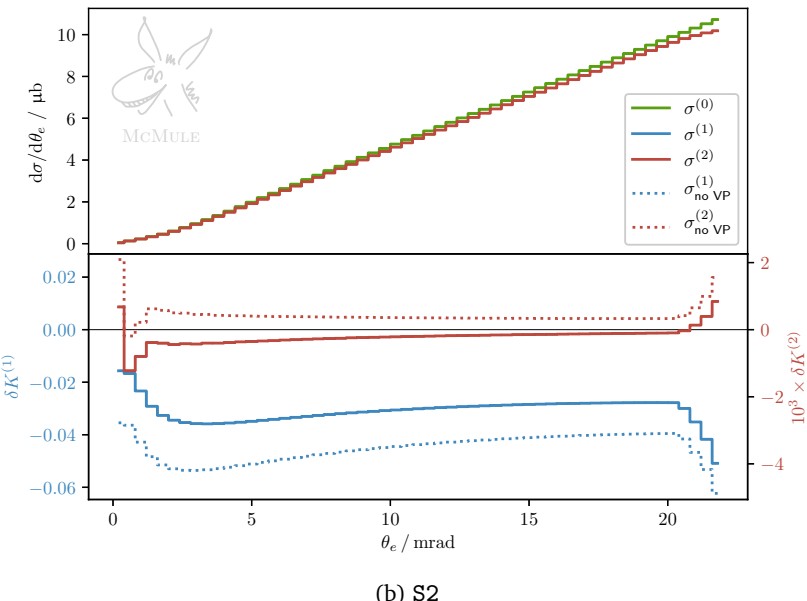

(b) S2

Figure 7: The differential cross section w.r.t. $\theta_e$ at LO (green) and NNLO (red) for scenarios S1 (without elasticity cut) and S2 (with elasticity cut). The NLO and NNLO $K$ factors are shown in blue and red, respectively.

Furthermore, sizeable NLO and NNLO corrections of up to 30% and 0.5%, respectively, can be observed. Naively, one could therefore conclude that the target precision of 10 ppm of MUonE is far out of reach. First of all, however, it has to be noted that the enhancement of the corrections at the end points of the distributions are due to soft photon emission. For a reliable description in this region, the logarithms need to be resummed. The leading logarithms can be resummed with a parton shower. Moreover as detailed in [55], also the calculation of the next-to-leading logarithms to all orders might be feasible. Secondly, the elasticity cut has

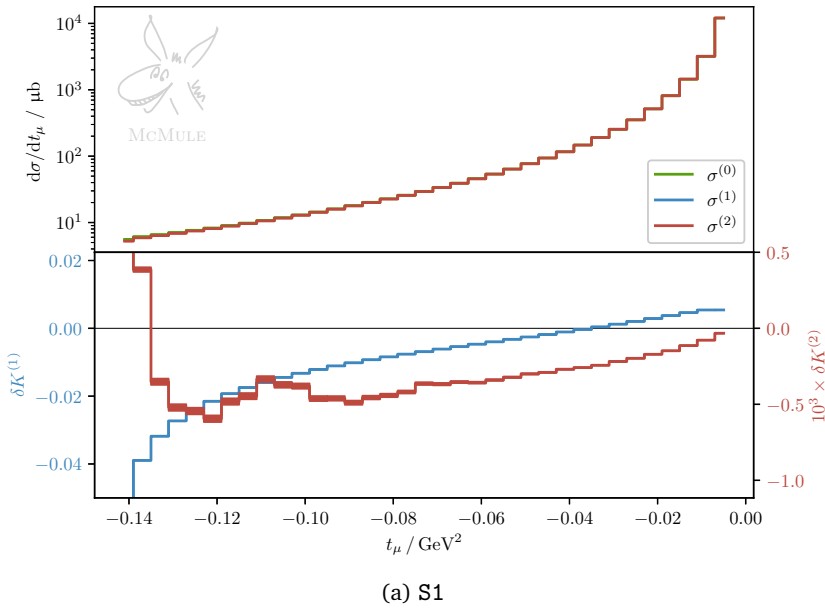

(a) S1

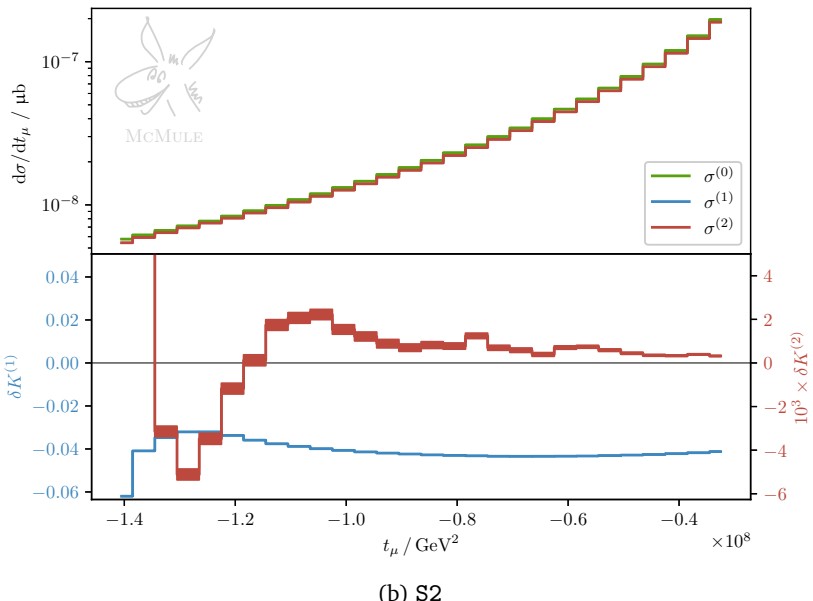

(b) S2

Figure 8: The differential cross section w.r.t. $t_\mu$ at LO (green) and NNLO (red) for scenarios S1 (without elasticity cut) and S2 (with elasticity cut). The NLO and NNLO $K$ factors are shown in blue and red, respectively.

the important effect of significantly reducing the variation in the $K$ factors. Since the MUonE experiment proposes to measure ratios of cross sections of different kinematic regions to cancel systematic uncertainties as opposed to absolute values, the flatness of the corrections is highly advantageous [55, 72].

# 6 Lepton-proton scattering with McMule

There is a long history in the study of elastic electron-proton and muon-proton scattering and the computation of radiative corrections to these processes started in the sixties [73]. If lepton-mass effects are taken into account appropriately, the two processes are the same from a computational point of view.

Typically, the corrections to lepton-proton scattering at NLO [74, 75] are split into different gauge-independent parts: corrections from the lepton line, corrections due to VP effects, corrections from the proton line and the so-called two-photon exchange corrections [76–80]. The latter contain contributions from inelastic intermediate states. This makes it difficult to obtain solid predictions from first principles. Looking at the charge asymmetry, i.e. the difference between $\ell^+ p$ and $\ell^- p$ scattering, is a useful tool to gain more information on two-photon exchange [81].

Going beyond NLO, the situation becomes considerably more complicated. To mention just two complications, the VP effects cannot be factorised any longer [57] and apart from the emission of photons also the emission of an additional $\ell^+\ell^-$ pair potentially needs to be considered.

Due to the small mass of the electron, corrections from the lepton line are typically dominant for $ep \to ep$. Hence, they received particular attention. Effects beyond the the soft approximation for the radiation from the electron were considered [82], as well as resummation of leading logarithmic effects [83]. Recently a calculation including the corresponding NNLO effects was presented in [84]. Resummation was also studied in an effective theory approach [85].

For these corrections, the only difference between lepton-proton scattering and the results presented in the previous section is due to the fact that the proton is not pointlike. This can be accounted for by parametrising the photon-proton interaction through form factors. Of course, electron-proton scattering has been used to determine these form factors and, in particular, their behaviour for small momentum transfer squared. This allows for an extraction of the proton radius, see e.g. [86]. However, for the results presented in this section we will simply use the standard dipole form factors, as given in Appendix A.

Thus, in this section we show NNLO results for unpolarised elastic lepton-proton scattering,

$$\ell(p_1)p(p_2) \to \ell(p_3)p(p_4) \left\{\gamma(p_5)\gamma(p_6)\right\}, \tag{15}$$

in the approximation that the lepton interacts with the proton through the exchange of a single photon with the standard dipole form factor. All lepton mass effects as well as leptonic and hadronic VP effects are taken into account. On the other hand, two (or more) photon exchange as well as radiation from the proton is neglected. We also make the assumption that there are no additional lepton pairs in the final state.

## 6.1 NNLO effects in elastic electron-proton scattering

As a first example, we consider $e^- p \to e^- p$ in a setting with kinematics adapted to the P2 experiment [6]. An incoming electron of energy $E = 155 \, \text{MeV}$ is scattering off a proton initially at rest. We consider scattering angles in the range $25° < \theta_e < 45°$. Following [84], we also apply a cut on the energy of the outgoing electron and require $E_e > 45 \, \text{MeV}$.

Starting with the total cross section (subject to the cuts above) we list the results in Table 3. Apart from listing the full NLO and NNLO corrections, $\sigma^{(1)}$ and $\sigma^{(2)}$, we also give separately the VP contribution (leptonic and hadronic) to the NLO and NNLO corrections. While the NLO corrections are rather large (about 5% with the VP contributing about 1%) the NNLO corrections are below 0.1%.

Table 3: Results for the integrated cross section for the P2 setting at LO, NLO, and NNLO.

|  | $\sigma/\mu\mathrm{b}$ | $\delta K^{(i)}/\%$ |
|---|---|---|
| $\sigma^{(0)}$ | 34.5392 | |
| $\sigma^{(1)}$ | 1.7763 | 5.1430 |
| $\sigma^{(1)}_{\mathrm{VP}}$ | 0.4663 | 1.3501 |
| $\sigma^{(2)}$ | -0.0237 | -0.0653 |
| $\sigma^{(2)}_{\mathrm{VP}}$ | 0.0132 | 0.0364 |
| $\sigma_2$ | 36.2919 | |

The first differential observable we consider is $d\sigma/d\theta_e$. In the top panel of Figure 9 we show the LO (green) and NNLO (red) differential cross section. The latter includes VP contributions. In order to assess the effect of higher-order corrections we show the $K$ factors in the bottom panel. The solid (dotted) lines refer to the corrections with (without) VP contributions. Since there are no large logarithms for this observable, the size of the corrections is in agreement with the expectation due to the counting of powers of $\alpha$ for all values of $\theta_e$. Consequently, the missing $N^3LO$ contributions due to emission from the electron are expected to be $\mathcal{O}(10^{-6})$ and, hence, negligible. Emission from the proton and two-photon exchange contributions, however, will need to be properly taken into account.

Results similar to those shown in Figure 9 have been presented in [84], not including VP contributions. Our NLO results (without VP) agree with these results. However, we disagree substantially with the NNLO corrections of [84], even if we adapt to their calculation and include the electron loop in the two-loop vertex diagram. In fact, our NNLO corrections are negative, whereas those presented in [84] are positive. With respect to the results presented in Section 5.2 that have been verified independently by [60], the only new ingredients are the matrix elements. They have been compared pointwise with [84] and agree.

As a second example we show $d\sigma/d|t|$ in Figure 10. The difference between the LO result (green) and NNLO result (red) is barely visible in the top panel. The size of the higher-order correction can be read off from the lower panel. While at LO, $t = (p_2-p_4)^2$ determined from the proton kinematics is the same as $t_e \equiv (p_1 - p_3)^2$ determined from electron kinematics, these two quantities start to differ at NLO. In our approximation the 'true' $Q^2$ that enters the form factors is $Q^2 = -t$. To illustrate the difference, we show the $K$ factors with $|t|$ as well as the electronic $|t_e|$. The size of the corrections differs by about 20% between the two observables.

Generally speaking, the corrections are well under control for most values of $Q^2 = |t|$, but increase towards the endpoint as in Figure 8. Indeed, NLO (NNLO) correction up to 10% (0.5%) are found in the tail of the distribution, and to obtain a very precise theoretical prediction in this region, large logarithms would have to be resummed.

## 6.2 NNLO effects in elastic muon-proton scattering

Elastic muon-proton scattering $\mu p \rightarrow \mu p$ can be used to obtain an independent extraction of the proton radius and shed light on possible differences between muons and electrons. In fact, MUSE [9] will measure simultaneously $\ell^\pm$-$p$ scattering with $\ell \in \{e, \mu\}$. Since we are neglecting two-photon exchange, there is no difference between $\ell^+$ and $\ell^-$ and the only difference to the process of Section 6.1 is the mass of the lepton. As we will see below, the larger mass of the muon typically results in smaller corrections.

For the purpose of illustration we consider an incoming muon of momentum $|\vec{p}_1| = 210\,\mathrm{MeV}$ scattering off a proton at rest. For the scattering angle range we use $20° < \theta_\mu < 100°$, as appropriate for MUSE. We include the same contributions as in Section 6.1. Again we start

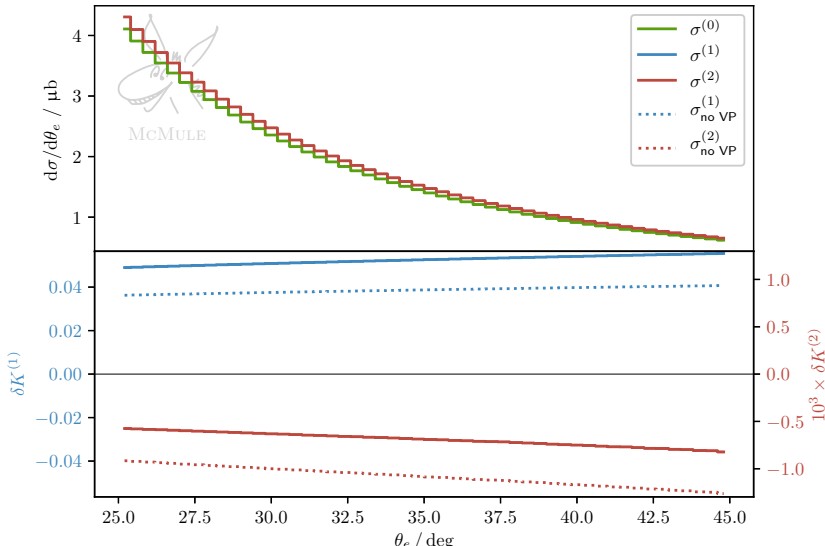

Figure 9: Differential cross section $d\sigma/d\theta_e$ for a P2 setting at LO (green) and NNLO (red) with $K$ factors. Solid (dotted) $K$ factors are with (without) the inclusion of VP contributions.

with the total cross section (subject to the cuts above) and present our results in Table 4. In this case, the NLO (NNLO) corrections are just over $10^{-2}$ ($10^{-4}$) and are actually dominated by the VP contributions.

The results for the differential cross section $d\sigma/d\theta_\mu$ are depicted in Figure 11. Again we show the $K$ factor with (solid) and without (dotted) VP contributions. This shows the dominance of the VP effects which themselves are entirely driven by the contribution of the electron. The corrections are roughly a factor 4 smaller than for electron-proton scattering shown in Figure 9. Accordingly, we expect $N^3$LO corrections from the emission of the muon to contribute well below $\mathcal{O}(10^{-6})$ to $d\sigma/d\theta_\mu$. This is encouraging in particular if these effects are seen as background to measure and study two-photon contributions.

As a second differential observable we consider $d\sigma/dE_\mu^{\text{kin}}$, where the kinetic energy of the muon is defined as $E_\mu^{\text{kin}} \equiv E_\mu - m_\mu$. At LO there is a one-to-one relation between the scattering angle $\theta_\mu$ and $E_\mu^{\text{kin}}$. Beyond LO, for a given $\theta_\mu$ there will be events with smaller $E_\mu^{\text{kin}}$ due to

Table 4: Results for the integrated cross section for the MUSE setting at LO, NLO, and NNLO.

|  | $\sigma/\mu$b | $\delta K^{(i)}/\%$ |
|---|---|---|
| $\sigma^{(0)}$ | 49.6677 | |
| $\sigma^{(1)}$ | 0.6541 | 1.3170 |
| $\sigma^{(1)}_{\text{VP}}$ | 0.7172 | 1.4440 |
| $\sigma^{(2)}$ | 0.0075 | 0.0150 |
| $\sigma^{(2)}_{\text{VP}}$ | 0.0076 | 0.0151 |
| $\sigma_2$ | 50.3294 | |

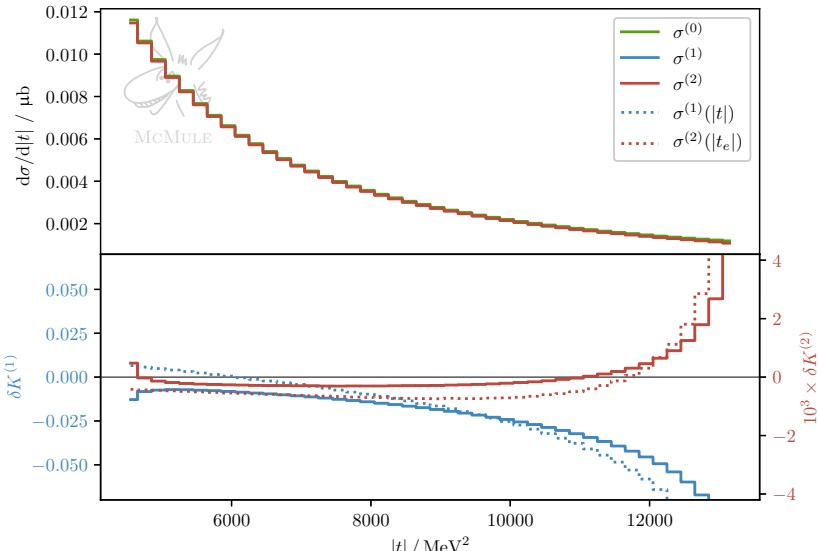

Figure 10: Differential cross section $d\sigma/d|t|$ for a P2 setting at LO (green) and NNLO (red) with $K$ factors. Solid (dotted) $K$ factors are determined from proton (electron) kinematics.

additional radiation. In order to illustrate this, we define four $\theta_\mu$ bands as follows:

$$
\begin{aligned}
\text{band 1}: \quad & 22.206° < \theta_\mu < 44.169° \quad & 126\,\text{MeV} > E_\mu^{\text{kin}}\big|_{\text{LO}} > 117\,\text{MeV} \\
\text{band 2}: \quad & 46.148° < \theta_\mu < 62.678° \quad & 116\,\text{MeV} > E_\mu^{\text{kin}}\big|_{\text{LO}} > 107\,\text{MeV} \\
\text{band 3}: \quad & 64.443° < \theta_\mu < 80.402° \quad & 106\,\text{MeV} > E_\mu^{\text{kin}}\big|_{\text{LO}} > 97\,\text{MeV} \\
\text{band 4}: \quad & 82.222° < \theta_\mu < 99.663° \quad & 96\,\text{MeV} > E_\mu^{\text{kin}}\big|_{\text{LO}} > 87\,\text{MeV}.
\end{aligned}
\tag{16}
$$

The corresponding values for $E_\mu^{\text{kin}}$ at LO are also indicated. At LO, all events of a given band will fall into this range of $E_\mu^{\text{kin}}$. This can be seen in the top panel of Figure 12, where $d\sigma/dE_\mu^{\text{kin}}$ at NNLO is shown in red (band 1), azure (band 2), green (band 3), and yellow (band 4). Outside the LO $E_\mu^{\text{kin}}$ range, the cross section falls sharply and is only non-zero due to radiative events. The middle panel shows the NLO $K$ factor. Since $K^{(1)}$ is formally infinity outside the LO $E_\mu^{\text{kin}}$ range, this factor is only shown in the region where the LO cross section does not vanish. Finally, in the lowest panel we show the NNLO $K$ factor. Within the LO $E_\mu^{\text{kin}}$ range, these corrections are small in accordance with the $\alpha^2$ suppression. Outside the LO $E_\mu^{\text{kin}}$ range, however, the NNLO corrections are quite large, up to 1.5%. This is not very surprising, since in this kinematic regime the NNLO terms are in fact only a NLO description of the observable.

## 7  Future developments of MCMULE

Once a mule has made up its mind, it is difficult to stop it. Hence, there will be continuous further developments and extensions of the code.

Roughly speaking, further developments can be divided into two classes. First, new processes or more complete descriptions of already implemented ones will be added. Second,

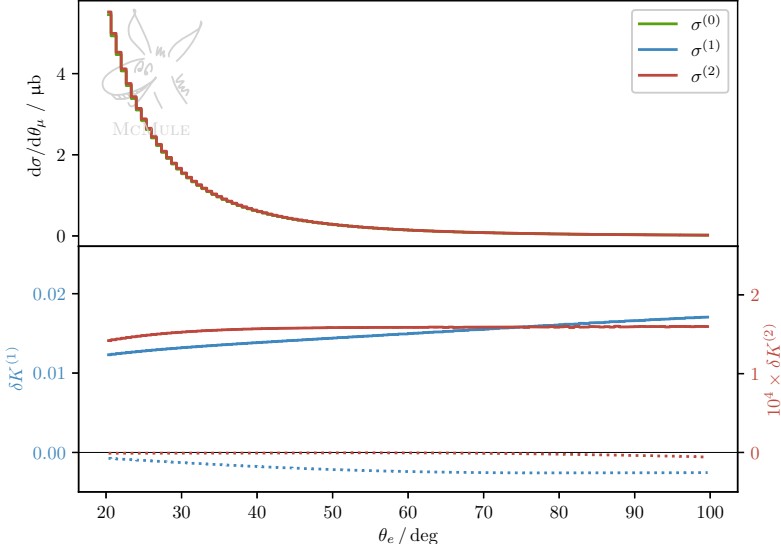

Figure 11: Differential cross section $d\sigma/d\theta_\mu$ for MUSE with incoming muons of momentum 210 MeV, at LO (green) and NNLO (red) with $K$ factors. Solid (dotted) $K$ factors are with (without) the inclusion of VP contributions.

there will be technical advances that improve the performance and precision, and potentially enable the implementation of more complicated processes.

With respect to the first class, work is ongoing to implement Møller scattering and $e^+e^- \to \gamma\gamma$ at NNLO. As an example for improvements on already implemented scattering processes we mention the inclusion of polarisation effects. So far, only the descriptions of decays are available for polarised initial states. Many of the processes mentioned in the introduction are, however, related to measurements of asymmetries that typically require polarised initial states. Such observables also often require the inclusion of electroweak corrections, as they lead to parity-violating effects. In addition, a full NNLO description of muon-electron scattering is envisaged. In order to go beyond the approximation of electronic corrections the full two-loop matrix element is required. The corresponding integrals in the limit of massless electrons are known [87,88] and the amplitude is being computed [89]. In addition, also the one-loop matrix element for $e\mu \to e\mu\gamma$ is required. The implementation of one-loop amplitudes for NNLO calculations requires particular care, since they are to be integrated over singular corners of the phase space. This results in two requirements. First, they have to be implemented with extreme numerical stability. Second, the numerical evaluation has to be reasonably fast.

To address these issues, in the long term it is probably advisable to link McMule to a dedicated code that evaluates higher-order amplitudes. There are several one-loop tools that specialise in this (for example [90–93]). While so far all attempts regarding automated computations of one- and two-loop amplitudes were dedicated to high-energy processes, it should be possible to adapt these tools to QED computations with massive fermions. OpenLoops [94] is one such tool that in the past has been relied upon for real virtual corrections. We also plan to set up an interface to OpenLoops to facilitate their numerically stable calculation. Of course, a major hurdle on the path towards using an external tool for all amplitudes is that the tool would have to be extended to two-loop calculations. While first steps have been made in this direction [95], we anticipate that two-loop amplitudes will have to be implemented directly in McMule for the time being.

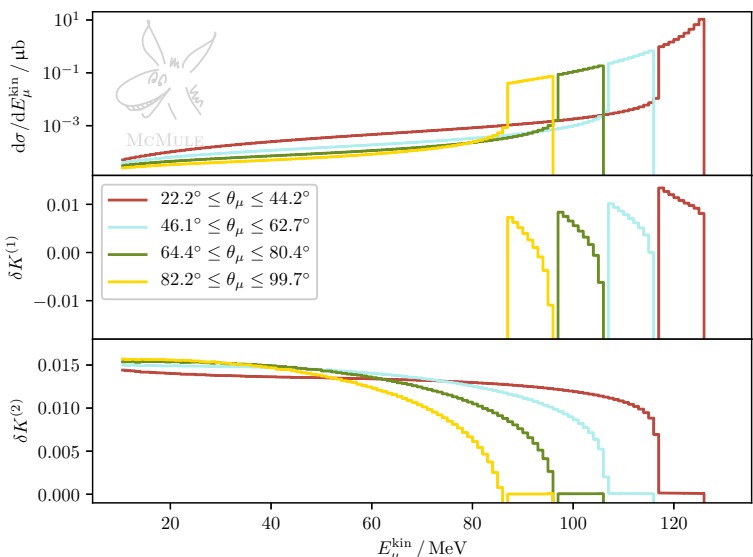

Figure 12: Differential cross section $d\sigma/dE_\mu^{\text{kin}}$ for MUSE with incoming muons of momentum 210 MeV, at LO (green) and NNLO (red) with $K$ factors. Results are shown separately for different bands of the scattering angle $\theta_\mu$.

Also related to the numerical stability of the integration is the treatment of pseudo singularities related to near collinear emission of photons. This is dealt with by splitting up the phase space such that only a small number (ideally one) of pseudo singularities is possible in each partition. Then the phase space is tuned such that there is a simple one-to-one match between the dangerous regions of phase space and an integration variable. Such a phase-space parametrisation typically results in a stable and reliable numerical evaluation of the integrals. As a possible further development, there is the option to subtract the pseudo-collinear singularity and add back a partially integrated counterterm [96]. However, since the logarithms arising from these phase-space region are physical, it is important to have a very flexible and exclusive treatment of the final-state particles.

Since FKS$^\ell$ works at all orders in perturbation theory, it is only the lack of the matrix elements that prevents us from going beyond NNLO. One example, where a N$^3$LO calculation might be feasible in the near future concerns the dominant electronic contribution to muon-electron scattering. As a more futuristic development we mention the idea to possibly compute the finite (eikonal-subtracted and ultraviolet-renormalised) matrix elements $\mathcal{M}_{n+i}^{(\ell-i)f}$ that are the ingredients of FKS$^\ell$ directly numerically.

Finally, many observables will be dominated by large logarithms, at least in some range of the distributions. Combining fixed-order calculations with a QED parton shower is a generic and powerful tool to resum the leading logarithms. Thus, the mule might want to take a shower after a hard day's work. The structure of FKS$^\ell$ is particularly amicable to a YFS parton shower because it already exploits the YFS structure. Initial (final) state collinear logarithms can be resummed using the parton distribution (fragmentation function) approach, which was recently extended to next-to-leading logarithmic accuracy [97].

Apart from technical developments we have also made steps towards being as open as possible with our results and facilitating their cross checks. All data that has been used in the plots presented here are available on a public git repository https://gitlab.com/mule-tools/user-library. For each data set, we give the input data and a SHA1 identifier of the code used

to create it. Since the code is available as a Docker image, anyone will be able to reproduce our results, regardless of operating system and dependencies. We hope this will accelerate progress in the theoretical description of low-energy particle physics experiments.

## Acknowledgements

We are grateful to M. Fael for help with implementing the vacuum polarisation contributions as well as providing a MathLink interface for `alphaQED`. In addition we thank G.M. Pruna for his contributions to the implementation of the muon-decay matrix elements. We also thank A. Gurgone and N. Schalch for their contributions to the further development of McMule. Moreover, we acknowledge useful discussions with A. Antognini, V. Ravindran, and M. Spira. It is a pleasure to thank C. Carloni Calame, M. Chiesa, S. Mehedi Hasan, G. Montagna, O. Nicrosini, and F. Piccinini for sharing their results for validation of our muon-electron scattering results, as well as R. Bucoveanu and H. Spiesberger for comparisons of electron-proton scattering. We are further grateful to E. Bagnaschi for his help integrating *udocker* into our workflow and to M. Zoller for validating some amplitudes with OpenLoops. TE and YU acknowledge support by the Swiss National Science Foundation (SNF) under contract 200021_178967. YU further acknowledges partial support by a Forschungskredit of the University of Zurich under contract number FK-19-087. PB acknowledges support by the European Union's Horizon 2020 research and innovation programme under the Marie Skłodowska-Curie grant agreement No 701647.

## A    Input parameters

The computations in this paper are performed in the on-shell scheme for the coupling and using pole masses. Accordingly, the input parameters we use are [98]

$$
\begin{aligned}
\alpha &= 1/137.035999084, & G_F &= 1.1663787 \cdot 10^{-11}\,\text{MeV}^{-2}, \\
m_e &= 0.510998950\,\text{MeV}, & m_\mu &= 105.658375\,\text{MeV}, \\
m_\tau &= 1776.86\,\text{MeV}, & m_p &= 938.272088\,\text{MeV}.
\end{aligned}
\tag{17}
$$

To convert from MeV to μb we use $(c\hbar)^2 = 1 = 3.89379372 \cdot 10^8\,\text{MeV}^2\mu\text{b}$. When presenting branching ratios of the muon, we always divide by the full width, determined from the lifetime $2.196981 \cdot 10^{-6}\,\text{s}$ as $\Gamma_\mu = 2.995984 \cdot 10^{-16}\,\text{MeV}$.

The interaction of the photon of momentum $q = p' - p$ with the proton is parametrised as

$$
\bar{u}(m_p, p')\Big(F_1(Q^2)\gamma^\mu + F_2(Q^2)\frac{i\sigma^{\mu\nu}q_\nu}{2m_p}\Big)u(m_p, p),
\tag{18}
$$

where $Q^2 = -q^2 \geq 0$. The form factors $F_1$ and $F_2$ are related to the Sachs form factors as

$$
G_E = F_1 - \tau F_2, \qquad\qquad G_M = F_1 + F_2,
\tag{19}
$$

where $\tau \equiv Q^2/(4m_p^2)$. Using the standard dipole parametrisation with $\Lambda^2 = 0.71\,\text{GeV}^2$ we set

$$
F_1(Q^2) = \frac{1 + \kappa\tau}{1 + \tau}\Big(1 + \frac{Q^2}{\Lambda^2}\Big)^{-2} \quad\text{and}\quad F_2(Q^2) = \frac{-1 + \kappa}{1 + \tau}\Big(1 + \frac{Q^2}{\Lambda^2}\Big)^{-2}.
\tag{20}
$$

Here $\kappa = 2.79284734$ is the proton's magnetic moment in units of the nuclear magneton.

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
