# Peer review of "QED at NNLO with McMule"

_SciPost Physics, doi:SciPost Phys. 9, 027 (2020)_

## Round 1 · Referee Report · Anonymous (Referee 1) · 2020-7-24

Strengths

1- The paper contains very important results needed for the interpretation of forthcoming high-precision experiments with lepton scattering 2- The paper is well written

Weaknesses

None

Report

The paper should be published.

Requested changes

1- In section 2, equations 2, 5, 6 the authors use the symbol \cal{M} and call this 'the matrix element', but for the calculation of cross sections one would need squares of matrix elements. 2- Some plots (e.g. in Fig.4, 6) or tables (e.g. Tab.2) do not exhibit units, and in Figs.8, 10, 12, the units are wrong. This should be corrected.

  • validity: high
  • significance: top
  • originality: high
  • clarity: top
  • formatting: excellent
  • grammar: excellent

Author:  Adrian Signer  on 2020-09-08  [id 944]

(in reply to Report 1 on 2020-07-24)

1) We have made our statement more precise just after Eq.(2.c) 2) We have added the units where there was an ambiguity, e.g. in Figure 6. We refrain from repeating the units of (e.g. the energy) on the y-axis if they have been given in the x-axis, e.g. in Figure 10 we do not write the full d\sigma/d|t| / (\mu b/MeV^2) . We think this is a reasonable and unambiguous short cut to state our units.

---

## Round 1 · Referee Report · Anonymous (Referee 2) · 2020-7-31

Report

This paper describes the new program McMule. It is a framework written in Fortran that performs high-order calculations in QED for scattering and decay processes involving leptons.

At the moment, the program implements muon decay, muon-electron scattering and proton-electron scattering. Depending on the process, it considers NLO, NNLO or dominant NNLO corrections, i.e., NNLO corrections only to the single photon exchange. The authors plan further developments of the existing code either by by including new processes or by improving the prediction for the existing ones.

Even if several of this processes were already computed in the past to various levels of precision, McMule represents the first comprehensive framework that includes all of them and that is fully differential, thus allowing studies with arbitrary cuts. The code is in a stable stage and it has been cross-checked with other independent calculations for all processes that are implemented.

The computational techniques used in McMule are briefly described in section 2. Sections 3 and 4 discuss the code's structure and how to run it. Sections 5 and 6 presents phenomenological results for muon-electron scattering, connected to the MUonE experiment, and proton-electron scattering, for P2 and MUSE experiments.

I think the work presented in this paper will have a large impact in particle physics in the following years and it will be strategic for future analysis in low-energy experiments like MUonE, P2, MUSE but also MEG and Mu3e. Therefore I am happy to recommend the manuscript for publication.

I want to make two minor suggestions to the authors. 1) In section 6.1 it is mentioned that their code substantially disagrees with the NNLO corrections in ref. [84]. I think they should better quantify it. Is it 1% or 100%? The reader might also wonder which experimental analyses employed the findings in [84] and therefore could be affected by them.

2) In section 2 it is briefly mentioned that if two-loop amplitudes are available only for massless external particles, "massification" can be used. However it is not explained what is is meant by that.

  • validity: high
  • significance: high
  • originality: high
  • clarity: top
  • formatting: excellent
  • grammar: perfect

Author:  Adrian Signer  on 2020-09-08  [id 945]

(in reply to Report 2 on 2020-07-31)

1) We have added a remark in Section 6.1 to make it clear that the disagreement is not small (i.e. more 100% than 1% ). Since the total cross section is affected, so are all distributions. 2) We have added a sentence in the first paragraph of Section 2 to say what "massification" is. The details of this procedure are described in the references given there.

---

## Editorial Decision

published